# Application of EN 16615 (4-Field Test) for the Evaluation of the Antimicrobial Activity of the Selected Commercial and Self-Made Disinfectant Wipes

**DOI:** 10.3390/ijerph18115932

**Published:** 2021-05-31

**Authors:** Stefan Tyski, Wanda Grzybowska, Ewa Bocian

**Affiliations:** 1Department of Antibiotics and Microbiology, National Medicines Institute, 00-725 Warsaw, Poland; wanda.grzybowska@op.pl (W.G.); e.bocian@nil.gov.pl (E.B.); 2Department of Pharmaceutical Microbiology, Medical University of Warsaw, 02-007 Warsaw, Poland

**Keywords:** biocidal efficacy, antimicrobial activity, disinfection, disinfectant agents, EN 16615 (4-field test), commercial disinfectant wipes, self-made wipes

## Abstract

The purpose of disinfectants is to reduce microorganisms on a contaminated surface and to prevent the spread of microorganisms. The relatively new EN 16615 simulates disinfection by wiping and allows for assessing the recovery of microorganisms from the surface and, importantly, the degree of spread of microorganisms when the surface is disinfected by wiping. For the first time, using this standard, the tested products in the form of commercial disinfectant wipes were compared with self-made wipes soaked in respective disinfectant liquids. The disinfected surfaces were simulated by homogeneous polyvinyl chloride plates. The studies were carried out not only with the standard, but also with clinical multidrug-resistant microbial strains. Based on the research, it can be concluded that the most effective products in the disinfection process (log_10_ reduction of ≥5) with the shortest contact time (1 min) were products containing ethanol, propanol, and quaternary ammonium compounds (self-made wipes) and propanol (commercial wipes). The least effective products (log_10_ reduction of <5) in terms of the contact time declared by the manufacturer were products containing ethanol and sodium hypochlorite (commercial wipes). Much better antimicrobial activity of self-made wipes was observed in comparison to the activity of the commercial wipes.

## 1. Introduction

Microbes contaminating inanimate surfaces in the patient zone (floors, countertops, beds, and medical equipment) constitute a reservoir of potential pathogens, including multi-resistant strains. They may play a role in the cross-transmission of pathogens and the subsequent colonization or infection in patients. Cleaning procedures, which are not always effective, allow bacteria resistant to commonly used disinfectants to survive in the environment and create a biofilm. Contamination of inanimate surfaces may occur as a result of the direct transfer of bacteria by the patient (higher from the infected patients than the colonized ones) or by the hands of health care professionals [1,2]. It is known that bacterial pathogens, both Gram-positive and Gram-negative bacteria, and fungal pathogens, such as *Candida albicans*, can survive for several months on dry environmental surfaces in hospitals and health care facilities despite regular cleaning and disinfection [3,4]. The longer a hospital strain remains on a surface, the longer it can pose a risk to a susceptible patient or health care worker. 

Health care environments, including frequently touched surfaces, must be regularly cleaned or cleaned and disinfected. Traditional methods of cleaning these areas, as well as non-contact methods of microbial decontamination (e.g., Ultraviolet light, hydrogen peroxide vapor), have limitations. The most commonly used agents for disinfecting the patient area are alcohols, quaternary ammonium compounds, chlorine compounds, and hydrogen peroxide. The purpose of the disinfectant product is to reduce microorganisms on the contaminated surface and to prevent the spread of microorganisms over the surface or their transfer from one surface to another [5].

Disinfectant efficacy testing is regulated by the Technical Committee No. 216 “Chemical Disinfectants and Antiseptic,” which is part of the European Committee for Standardization (CEN).

Two study phases were developed to evaluate the effect of disinfection. The areas of application of disinfectants in both phases are human medicine, veterinary use, food hygiene, and domestic and institutional use. Both phases include testing the bactericidal, fungicidal, and sporicidal activity, and, in addition, phase 2 involves testing the tuberculocidal and virucidal activity of the disinfectants. In phase 2, which includes steps 1 and 2, an organic or inorganic load is introduced to demonstrate whether the test product reacts with other substances, such as proteins. Phase 2, step 2 is a carrier method that uses contaminated surfaces, such as metal discs, glass plates, polyvinyl plates, or the surface of the hands. The standards listed above help to evaluate the overall antimicrobial efficacy of the product but do not distinguish the mechanical removal of contamination from the chemical inactivation of test organisms. The EN 16615 standard, which was relatively recently developed, simulates disinfection by wiping and allows for assessing the recovery of microorganisms from the surface and, importantly, the degree of spread of microorganisms when the surface is disinfected by wiping. Wipes soaked in disinfectants are used for cleaning and disinfecting surfaces of different medical devices and equipment, e.g., ultrasound heads, surfaces of beds, treatment chairs, and operating tables, as well as fragments of floors and walls, if necessary. Moreover, the wiping process itself should be treated as one of the factors that may affect the effectiveness of disinfection [6]. Although the above standard EN 16615 concerns the testing of bactericidal and yeasticidal activity against standard microorganisms, there have been reports of the use of this 4-field method to determine biocidal activity against additional microorganisms, such as *Clostridium difficile* spores [7] or viruses, including SARS-CoV-2 [8,9]. 

In addition to the European manual method of testing the effectiveness of the disinfection process, the EN 16615 method, in the US, the automatic Wiperator method ASTM Standard E2967-15 has been established [10]. A comparison of these two methods can be found in the publication by Jacobshagen et al. [11]. The manual method better reflects the practical conditions of disinfection, while the automatic method, due to its nature, is better controlled.

The aim of our work was the practical application of a relatively new European standard dedicated for testing disinfectants using a carrier in the form of PVC plates and with the mechanical action. The methods used so far did not allow for testing products such as commercially available, ready-made wipes or self-soaked wipes under simulated conditions of their use.

## 2. Materials and Methods

### 2.1. Disinfectant Products

The active substances contained in these products belong to various chemical groups, such as alcohols, quaternary ammonium compounds, oxidizing compounds, and glucoprotamine. In total, 12 commercial products for disinfection were selected for the study, including 6 products in the form of a liquid ready for soaking wipes or in the form of a concentrate for the preparation of working solutions and 6 products in the form of commercial wipes soaked in a disinfectant (Table 1).

Products in the form of concentrates (No. 1, 2, and 4) were diluted in hard water containing 6 mL/L of solution A (19.84 g/L of magnesium chloride solution (Sigma) and 46.24 g/L of calcium chloride (Chempur, Karlsruhe, Germany)) and 8 mL/L of solution B (35.02 g/L of sodium bicarbonate (Chempur)).

All commercial wipes still had a valid shelf life, and their original packaging was opened just prior to testing. Self-made wipes soaked in disinfectant according to EN 16615 were used just after preparation.

### 2.2. Microbial Strains

The studies were carried out with the use of standard and clinical microbial strains. Bacterial strains from the American Type Culture Collections (ATCC): *Staphylococcus aureus* ATCC 6538, *Pseudomonas aeruginosa* ATCC 15442, *Enterococcus hirae* ATCC 10541 and as a representative of yeast—*Candida albicans* ATCC 10231, were used. The multi-resistant clinical strains, such as methicillin-resistant *Staphylococcus aureus* MRSA A876, methicillin-resistant *Staphylococcus epidermidis* MRSE 841, *Escherichia coli* 1043-producing extended-spectrum beta-lactamases (ESBL), and *Pseudomonas aeruginosa* 852 (also ESBL +), were also used in the study. These strains originated from the Clinical Hospital in Warsaw and the Department of Pharmaceutical Microbiology, Medical University of Warsaw.

### 2.3. Application of Standard EN 16615

The tests were carried out in accordance with the standard EN 16615:2015-06 “Chemical disinfectants and antiseptics. Quantitative test method for the evaluation of bactericidal and yeasticidal activity on non-porous surfaces with mechanical action employing wipes in the medical area (4-field test). Test method and requirements (phase 2, step 2)” [12]. This is a carrier method in which disinfected surfaces are simulated by homogeneous polyvinyl chloride (PVC) plates “Special 43 Plus E” (Tarkett, Jasło, Poland). Four fields of 25 cm^2^ each were marked on the plates in order to assess the ability to reduce the microbe count as well as the ability to spread microbes to subsequent areas of the surface. Test organism suspensions with a density of 1.5–5.0 × 10^9^ cfu/mL (bacteria) and 1.5–5.0 × 10^8^ cfu/mL (*C. albicans*), were estimated by DEN-1B McFarland Densitometer, Grant Instruments Ltd., England, in the presence of a 3 g/L bovine albumin solution (Sigma, St. Louis, MO, USA), and 3 mL/L of sheep erythrocytes (Grasso, Starogard Gdański, Poland) simulating dirty conditions were applied in an amount of 0.05 mL and spread over the first field of the test surface. The test surface was dried at room temperature, which caused the microorganisms to attach to the plates. Then, a granite block weighing approximately 2.5 kg, providing adequate pressure on the surface, was covered with a 17 cm × 30 cm wipe made of low-dusting non-woven fabric (TORK, SCA, Göteborg, Sweden) soaked in 16 mL of the test disinfectant (products 1–6) or with a commercial wipe (products 7–12). Such prepared unitary weight with the soaked wipe was used for the wiping process on the surface contaminated with microorganisms beginning from 1 field up to 4 and back down to 1 field within 2 s. The wipes were weighed just before and immediately after use to determine the amount of disinfecting liquid released from the wipe (Figure 1).

After a chosen contact time, the activity of the product was neutralized at a temperature of 20 ± 1 °C with an appropriately selected neutralizer for 15 s or (5 min for a contact time of ≥15 min). For products containing alcohols, quaternary ammonium compounds, and a chlorine compound, the Dey/Engley neutralizer (Becton Dickinson, Sparks, MD, USA) of the following composition was used: 5.0 g/L pancreatic casein extract, 2.5 g/L yeast extract, 10.0 g/L dextrose, 1.0 g/L sodium thioglycolate, 6.0 g/L sodium thiosulphate, 2.5 g/L sodium bisulphite, 805.0 g/L polysorbate, 7.0 g/L lecithin, and 0.02 g/L bromocresol purple. For the product containing hydrogen peroxide, 0.25 g/L catalase solution (Sigma, St. Louis, MO, USA) was used. The product containing glucoprotamine was neutralized with a solution of the following composition in a diluent (tryptone, 1.0 g/L pancreatic casein extract (Becton Dickinson, Sparks, MD, USA), and 8.5 g/L sodium chloride (POCH, Gliwice, Poland)): 3 g/L lecithin (AppliChem, Darmstadt, Germany), 8030 g/L polysorbate (POCH, Gliwice, Poland), 5 g/L sodium thiosulfate (Honeywell, Muskegon, MI, USA), 1 g/L L-histidine (Merck, Darmstadt, Germany), and 30 g/L saponin (AppliChem, Darmstadt, Germany).

After the wiping process was complete, the recovery procedure with the swabs was performed from each field on the test surface. The swabs were rinsed in a suitable neutralizer (5 mL), and 1 mL aliquots were inoculated in duplicate using the pour plate technique on the appropriate culture medium: trypticase soy agar (bioMerieux, Marcy l’Etoile, France) for bacteria and malt extract agar (Merck, Darmstadt, Germany) for *C. albicans*. The plates were incubated at 37 ± 1 °C (bacteria) for 20 to 24 h and 30 ± 1 °C (*C. albicans*) for 40 to 48 h. The colony-forming units were counted with the usage of Colony Counter apparatus, IUL Instruments, Spain.

The disinfectant product meets the requirements of PN-EN 16615 when the decimal log reduction of test microorganisms in the first field is at least 5 logs (bacteria) and 4 logs (yeast) and the average number of cells in the fields from 2 to 4 does not exceed 50 cfu. The following controls and validations were performed concurrently with the product activity test: control with water supplemented in 0.1% polysorbate 80 used instead of the product test solution to ensure that the test organisms are spread on the 4 fields and cells’ number reaches a certain level; drying control at time 0, just after drying (D_c0_), drying control after drying time, and exposure time (D_ct_) to ensure that the test organisms can survive in the test conditions, and control of neutralizer toxicity and validation of the dilution-neutralization method, to ensure that used neutralizer is suitable.

Experiments were performed three times. The arithmetic mean value and standard deviation were calculated.

## 3. Results

Products containing various active substances, such as alcohols, quaternary ammonium compounds, oxidizing compounds, and glucoprotamine in the form of liquids (products 1–6), which were used to soak wipes or in the form of commercial wipes (products 7–12), were used in the research (Table 1).

In parallel with product activity testing, controls with water used in place of the test product, controls of drying in time 0 (just after drying, D_co_), and controls drying after time including drying and exposure time (D_ct_) were run. For all tested microorganisms, with the exception of yeast, the above controls met the requirements of the standard.

In the case of the self-made wipes, the required reduction in bacteria counts, after the contact time declared by the manufacturer, from 1 min for alcohols to 15 min for glucoprotamine, was achieved for most of the tested products (Figure 2).

Only the product containing didecidimethylammonium chloride and ethylenediaminetetraacetic acid (No. 3) after the declared contact time of 10 min did not meet the requirements of the standard when three bacterial strains were used: *S. aureus* ATCC 6538 and clinical strains MRSE 841 and *E. coli* 1043. In the case of the standard strain, an additional study was carried out by extending the contact time. To obtain a log_10_ reduction above 5, it was necessary to extend the contact time up to 15 min.

The results obtained with the commercial wipes are more variable than those obtained with the self-made wipes. The wipes soaked in isopropyl alcohol (No. 9) showed a log_10_ reduction of over 5, which is required by the standard for all microorganisms tested. Within the declared contact time of 1 min, the wipes soaked with the mixture of quaternary ammonium compounds (No. 10) reduced all microorganisms tested except *E. coli* 1043, according to the standard. The wipes soaked in hydrogen peroxide (No. 12) showed activity against the tested bacterial microorganisms after the declared contact time of 2 min, with the exception of *P. aeruginosa* strains. Commercial wipes soaked in ethanol (No. 7) and sodium hypochlorite (No. 11) did not meet the standard requirements in the case of all tested bacteria after the declared time of 1 min.

For wipes soaked in sodium hypochlorite, the extended contact time allowed for an appropriate degree of reduction in the case of standard strains *S. aureus* ATCC 6538 (after 15 min) and *P. aeruginosa* ATCC 15442 (after 5 min), while in the case of the *E. hirae* ATCC 10541 strain even an extension of up to 15 min did not provide the expected reduction in cells. Similarly, in the case of wipes soaked in ethanol, extending the contact time to even 15 min did not result in a reduction in the number of microorganisms above 5 logs.

In parallel with the determination of the degree of reduction based on the number of colonies in the first field of the studied area, the number of colonies in the remaining fields (2–4) was determined. The disinfecting product meets the requirements of EN 16615 when the average number of colony-forming units in fields 2—4 does not exceed 50. In the case of the bactericidal products used at the appropriate concentration and contact time, the number of colonies did not exceed 50 cfu in fields 2–4.

In the case of products not showing an appropriate degree of reduction in the first field (ethanol and sodium hypochlorite), in fields 2–4 the microorganisms were both found or not found. The was an exceptionally high presence of bacterial cells in fields 2–4 from all tested microorganisms in wipes soaked in sodium hypochlorite (>1650 cfu per field). In the case of commercial wipes soaked with ethanol, a slight presence of bacteria was found for the tested microorganisms MRSE 841 (28 cfu per field), *P. aeruginosa* 852 (4 cfu per field), and the complete absence of bacterial cells in fields 2–4 was found in the case of the remaining microorganisms (*S. aureus* ATCC 6538, *E. hirae* ATCC 10541, *P. aeruginosa* ATCC 15442, *E. coli* 1043, and MRSA A876).

Bactericidal activity of substances from the group of alcohols (ethanol and isopropanol) and quaternary ammonium compounds (a mixture of didecidimethylammonium and alkylbenzyldimethylammonium chlorides), in both commercial and self-made wipes, were compared after the contact times declared by the manufacturers (Figure 3).

Products in both forms of wipes containing propanol showed bactericidal activity against the tested bacteria. In the case of products with ethanol, commercial wipes, unlike self-made wipes, did not show the required degree of bacteria count reduction against all tested strains. In the case of products containing quaternary ammonium compounds, both types of wipes reduced the number of cells according to the standard, except for the commercial wipes for the *E. coli* 1043 strain.

To explain the differences in the activity of the same active substances used in the form of commercial wipes and self-made wipes, each wipe was weighed before and after surface disinfection. The amount of disinfectant released from the self-made wipes (soaked with 16 mL of liquid) was 1.36 ± 0.17 g, and the average weight lost for the commercial wipes was 0.33 ± 0.05 g (except for product 9 (wipes soaked in isopropyl alcohol), where the average amount of the disinfectant agent released from the wipes was 0.72 g).

The cell count reduction of *C. albicans* was above 3 logs for all tested products (according to the EN 16615 required reduction of 4 logs). Nevertheless, the low recovery of *C. albicans* cells from the dried surface (D_co_ control, D_ct_ control, and water control) did not allow for the correct level of reduction. It may be possible that *Candida* cells do not sufficiently adhere permanently to the PVC surface and that cells are removed during the process of rubbing the surface. Such a situation makes results difficult to interpret. The logarithm of the cfu number during time 0 (D_co_), as well as the total drying time and contact time (D_ct_) for *C. albicans*, should be between 5.88 and 7.40 logs. In our study, despite the correct initial density for the *C. albicans* suspension that was 7.32–7.40, the values for D_co_ ranged from 5.58 to 5.97, and for D_ct_ the values were 5.40–5.78. By comparing the number of *C. albicans* cells in the suspension before drying and in the drying control at time t = 0 (just after drying), we observed a decrease in the number of cells log 1.51–1.75, similar to the observation of Chojecka (1.21–2.07) in studies assessing the survival of *C. albicans* on PVC surfaces [13]. Similarly, the decrease in the number of cells in the D_co_ and D_ct_ drying controls in our study was 0.13–0.19 compared to Chojecka’s results (0.03–0.37) [13].

Simultaneously, with the testing of biocidal activity of the disinfectants, a control using water instead of the product was performed. In the controls with water, the largest number of Gram-positive bacteria (*S. aureus*, *E. hirae*, MRSA, and MRSE) and the least number of Gram-negative bacteria (*P. aeruginosa* and *E. coli*) remained on the plate. In the case of yeast, the recovery of cells from the plate surface was below the standard (less than 10 cfu).

## 4. Discussion

The standards used for testing the antimicrobial activity of antiseptic and disinfectant products describe the phase 2, step 2 carrier methods using glass and metal surfaces or surface of hands. The EN 16615 standard, developed relatively recently, extends the application of disinfectants to new surfaces: polyvinyl plates. Moreover, this method simulates disinfection by mechanical action—wiping—and for assessing the recovery of microorganisms from the surface and the degree of spread of microorganisms when the surface is disinfected by wiping.

Disinfectants in the form of wiping or spraying solutions and commercial wipes soaked with a disinfectant are used to disinfect surfaces in medical facilities. Decontamination of surfaces takes place not only through the biocidal action of the disinfectant, but also through mechanical action—wiping. Commercial wipes are convenient for the user: they do not leave any wet residues on the surface and are easy to apply in a relatively short time. The question is whether the amount of disinfectant soaked in the commercial wipes is sufficient to achieve a biocidal effect, and whether the wipe time is sufficient to achieve the desired sanitizing effect. In the case of products used for spraying, manufacturers recommend a certain contact time of the product with the surface before starting the wiping process or allowing the product to dry, which may result in greater effectiveness. For both methods of applying the disinfectant on the contaminated surface, it is important to reduce the pathogen load to the lowest possible level and to avoid the spread of the microorganism over a larger area [14]. The method described in the standard EN 16615 also allows for checking the degree of spread of microorganisms on the tested surface during wiping. The presence of microorganisms in fields 1–4, despite disinfectant use, testifies to the lack of microbial reduction and possibility of spread during wiping. The absence of microorganisms in fields 2–4, despite their presence in field 1, in the case of wipes soaked in 70% alcohols, confirmed the possibility of the ‘fixation’ of microorganisms in the first field. Additionally, the test with the use of 30% alcohol (a concentration that does not kill bacteria) was carried out, and the spread of bacteria was obtained in all tested fields.

The difference in wipe weight before and after the wiping process determines the amount of substance released from the wipe. We observed greater loss of weight for the self-made wipes in comparison to the commercial wipe weight. This may explain the differences in the activity of both tested wipes (commercial and self-made wipes). The chemical group of the active substance is also important. Alcohols easily evaporate from the surface and with a short time of contact they do not show the effect of prolonged action. Another reason for the weaker activity of the commercial wipes may be the material from which they were made, the compatibility of the disinfectant with the wipe material, the use of too little disinfecting agent for the wipe, the way the product was packaged, and the presence or absence of substances that reduce surface tension, which improve wetting of the wiping surface, etc. [2,15].

Standard EN 16615 also concerns the quantitative assessment of the biocidal activity of disinfectants against yeast. Testing the activity using the *C. albicans* strain is a significant problem related to the viability of yeast cells on surfaces. According to the standard EN 16615, the average number of cfu in fields 2–4 in the control with water should not be less than 10, which may suggest retaining microorganisms on the used wipes and preventing their transfer to the surface. In the case of yeast, the recovery of cells from the plate surface was below the standard, which may indicate poor survival (sensitivity to drying) of *C. albicans*. The cfu values in fields 2–4 obtained in the control with water proved that wipes soaked in water retain microorganisms in an amount that meets the requirements of the standard for disinfection products. Gemein et al. performed an experiment with the application of an alternative plate material—PVC plate-free foam (Forex classic)-simulating disinfected surfaces, instead of PVC with polyurethane (PUR) [16]. The studies were carried out with the *S. aureus* strain; however, this change may allow for better survival and recovery of the *C. albicans* strain.

The choice of the tested disinfection products was dictated by the fact that they are most often used in the medical field. Alcohol enables effective surface wetting and is bactericidal and virucidal. It has no sporicidal activity. Alcohol-based disinfectants do not present significant toxicity problems but are highly flammable, corrosive to metals, and can cause the swelling and hardening of rubber and some types of plastics. Disinfectants with alcohol are suitable for disinfecting small surfaces, but due to their high volatility, it is difficult to ensure sufficient contact time in open systems.

In the conducted research, wipes soaked in ethanol (self-made wipes) had bactericidal activity in contrast to commercial wipes with ethanol. The lack of sufficient bactericidal activity was also demonstrated in the studies by Andersen et al. [17] and Tarka et al. [18]. No differences in the degree of microbial reduction were observed for the commercial wipes and for the wipes soaked in propanol.

The most commonly used halogen disinfectant is sodium hypochlorite due to its low cost, quick mode of action, and large bactericidal spectrum. Contrary to alcohols, it provides a prolonged bactericidal effect, but it destroys metal surfaces, can be inactivated by organic substances, and, with prolonged use, causes an irritating effect on the skin, eyes, and mucous membranes. In our research, commercial wipes soaked with sodium hypochlorite did not show required bactericidal activity by the standard EN 16615 after the contact time declared by the manufacturer. Lack of microbial reduction at the level required by the EN 16615 (log10 reduction of ≥5) after using wipes soaked in sodium hypochlorite may be due to the following reasons: the presence of high concentration of organic substances (3 g/L bovine albumin solution and 3 mL/L of sheep erythrocytes), the relatively small amount of soaking liquid remaining on the disinfected surface (0.40 ± 0.15 g), and a short contact time—1 min, declared by the manufacturer.

Oxidizing agents, such as chlorine compounds or hydrogen peroxide, can be used on large surfaces. Hydrogen peroxide has a bactericidal and sporicidal effect. This agent is relatively environmentally friendly due to its rapid degradation. The commercial wipes soaked in hydrogen peroxide that we tested in the contact time recommended by the manufacturer (2 min), did not show bactericidal activity in accordance with the standard EN 16615, only against *Pseudomonas* genus. Häring et al. obtained similar results (log_10_ reduction of >5) for enterococci using wipes soaked in 1.5% hydrogen peroxide with a contact time of 2 min [19]. 

Quaternary ammonium compounds are the most widely used disinfectants with a broad spectrum of biocidal activity. Even very low concentrations damage the cytoplasmic membrane of microorganisms. They are classified as low toxic risk products. They are effective against bacterial biofilm but are characterized by low effectiveness against Gram-negative bacteria and non-enveloped viruses. Benzalkonium chloride can cause severe inflammatory irritation through both contact and inhalation. Quaternary ammonium compounds are often used disinfectants in wipes; however, the adsorption of these compounds on the cotton material of the wipes can lead to disinfection failure [6]. In our study, quaternary ammonium compounds were effective against the tested bacteria in the case of both self-made and commercial wipes; however, their activity did not comply with EN 16615 against *E. coli* 1043. The reason for the lack of the bactericidal activity required by the EN 16615 standard against the clinical strain of *E. coli* 1043 may be explained by lower activity of QAC against Gram-negative bacteria than against Gram-positive bacteria [6]. The adsorption of some amount of active substance by the wipe material may be also possible, as was described by Tarka et al. [20]. The lack of the activity of CW wipes soaked with a mixture of QAC and EDTA (No. 3) required by the EN 16615 standard against three out of seven tested strains can be justified by the possibility of interaction of the active substance with the tissue material and the presence of organic substances.

Until recently, aldehydes and phenolic compounds were among the main components of disinfectants, but today, due to their toxicity, they are not used for routine surface disinfection. In the mid-1980s, an agent called glucoprotamine, which is an amine derivative, was introduced to the market and, unlike quaternary ammonium compounds, is effective against mycobacteria and a greater number of non-enveloped and enveloped viruses. Glucoprotamine is easily and quickly biodegradable, so it is safe for the environment. Metals and most synthetic substances are resistant to glucoprotamine, so it is safe for disinfecting tools and surfaces. The tested liquid containing glucoprotamine used to soak the wipes showed bactericidal activity against all tested microorganisms, according to the standard. Similar results (log_10_ reduction of >5) for enterococci were obtained by Häring et al. using an even lower concentration of glucoprotamine (0.5%) and shorter contact time (2 min) [19].

Based on the conducted research, it can be concluded that the most effective products in the disinfection process (log reduction of >5) and with the shortest contact time (1 min) were products containing ethanol, propanol, and quaternary ammonium compounds (self-made wipes) and propanol (commercial wipes). The least effective (log reduction of <5) in terms of the contact time declared by the manufacturer (1 min) were products containing ethanol and sodium hypochlorite (commercial wipes). Better activity of the self-made wipes was observed in comparison to the activity of the commercial wipes.

## 5. Conclusions

The EN 16615 standard can be used to test the antimicrobial activity of both commercial and self-made wipes, and not only is the biocidal potency of wipes soaked in liquids taken into account, but also the mechanical wiping to prevent the transmission of microorganisms. Self-made wipes eliminated microorganisms more efficiently than commercial wipes, which indicates the need for producers to used tighter packaging for wipes and to shorten the shelf life of the original sealed packaging. The differences in the bactericidal activity of SMW and CW wipes may be also due to the amount of soaking fluid. Wipe producers should use appropriate disinfecting liquids in preparations that allow for effective surface disinfection in accordance with the EN 16615. 

The assessment of the suitability of EN 16615 for determining the biocidal activity of disinfection products with the use of the *C. albicans* strain requires further research by the authors of the standard considering, for example, the modification or change of the carrier surface simulating the conditions of product use. Adherence of *Candida* cells to the PVC surface is much worse than adherence of bacterial cells. The method described in the EN 16615 standard, due to the low recovery of *C. albicans* cells from the dried surface, does not allow for proper testing of the yeasticidal activity.

## Figures and Tables

**Figure 1 ijerph-18-05932-f001:**
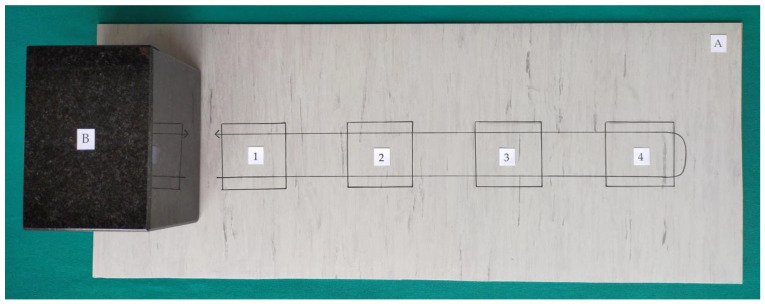
Model of the markings the four test fields on the test surface. (**A**) Test surface; (**B**) Unitary weight, 1–4: Test fields 25 cm^2^, 1: Inoculated field.

**Figure 2 ijerph-18-05932-f002:**
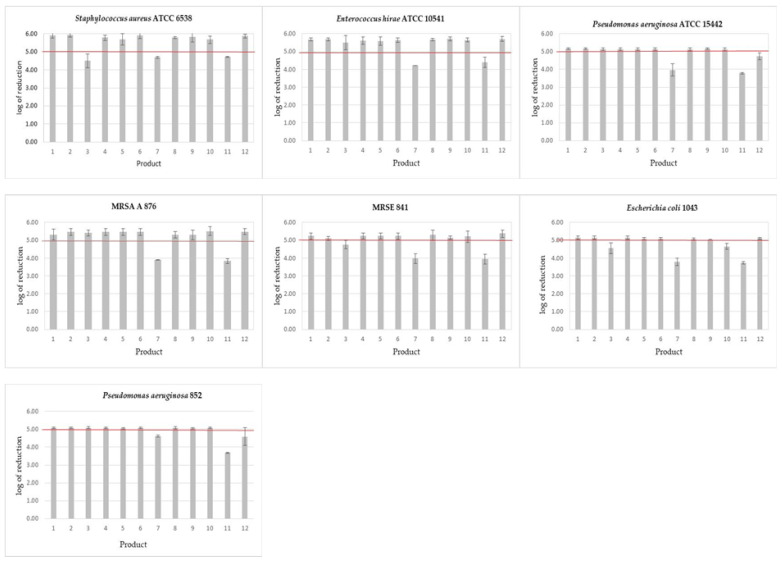
Bactericidal activity of disinfectant products against bacterial strains after the contact time declared by the manufacturers. Products 1–12—see Table 1; The results above the horizontal line (log 5) indicate results complying the EN 16615. Mean and SD values are presented.

**Figure 3 ijerph-18-05932-f003:**
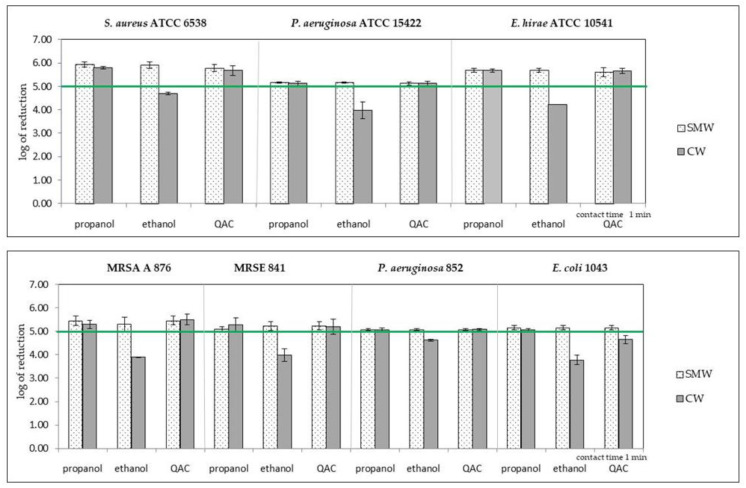
Comparison of bactericidal activity of self-made wipes (SMW) and commercial wipes (CW) soaked in three disinfectants for standard and clinical isolates. The results above the horizontal line (log 5) indicate results complying the EN 16615. Propanol: self-made wipes no. 2, commercial wipes no. 8; ethanol: self-made wipes no. 1, commercial wipes no. 7; QACs—quaternary ammonium compounds: self-made wipes no. 4, commercial wipes no. 10. Mean and SD values are presented.

**Table 1 ijerph-18-05932-t001:** Commercial products for disinfection and conditions of use recommended by the manufacturers.

No.	Content of Active Substances in 100 g of Product	Recomended Concentration of Product	Contact Time
**Liquids Used for Soaking Wipes**			
1	ethanol 70%	100%	not given
2	propan-2-ol 35 g propan-1-ol 25 g	100%	acc. VAH 5 min acc. NIPH-NIH 1min
3	didecyldimethylammonium chloride EDTA *no quantitative data*	1.6%	10 min
4	didecyldimethylammonium chloride 0.25 g, alkilbenzyldimethylammonium chloride 0.50 g	100%	1 min
5	didecyldimethylammonium chloride 5.9 g N-(3-aminopropyl)-N-dodecylpropane-1,3-diamine 7.4 g	1% 0.75% 0.5%	5 min 15 min 30 min
6	glucoprotamine 25 g	2.5% 4.0%	30 min 15 min
**Commercial Wipes Soaked in:**			
7	ethanol 70 g	---	1 min
8	propan-2-ol 35 g propan-1-ol 25 g	---	1 min
9	isopropyl alcohol 70 mL	---	5 min
10	alkilbenzyldimethylammonium chloride 0.50 g didecyldimethylammonium chloride 0.25 g	---	1 min
11	sodium hypochlorite > 5400 ppm	---	1 min
12	hydrogen peroxide 1.5 g	---	2 min-bacteria 1 min-yeast

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
