# Peer review of "Application of EN 16615 (4-Field Test) for the Evaluation of the Antimicrobial Activity of the Selected Commercial and Self-Made Disinfectant Wipes"

_ijerph, 2021, doi:10.3390/ijerph18115932_

Round 1

Reviewer 1 Report

The manuscript ijerph-1197538 of Tyski et al. entitled “Application of EN 16615 (4-Field Test) for the Evaluation of the Antimicrobial Activity of the Selected Commercial and Self Made Disinfectant Wipes” evaluated the bactericidal and yeasticidal activity of different disinfectants on polyvinyl chloride (PVC) plates using mechanical action with self-made wipes and commercial wipes. This work is interesting for readers of the International Journal of Environmental Research and Public Health. However, I believe that the scientific relevance of this article is not adequately justified. In addition, the promising and hopeful results reported by the authors in the abstract and conclusions section are not consistent with the results shown in the Results section. I find it to contain very few results for an Original Article and I suggest it for a Short Communication. The manuscript needs major revisions.

The manuscript is well written but some important information are missing or should be added. Authors should consider the following suggestions point by point to improve the manuscript:

The abstract should clearly indicate the novelty of the work.

In the Introduction section you should indicate some examples of medical equipments which this wipes can be applied and in which hospital equipment or surface PVC would be present.

Line 35: The reference used is from 1991 and a more current one should be used.

In line 39 it says “Candida albicans, can survive for several months on dry environmental surfaces in hospitals and health care facilities despite cleaning and disinfection”. This does not agree with the results obtained for Candida albicans in this study. The authors have not been able to evaluate the yeasticidal activity, justifying that the survival in PVC was low due to the sensitivity it shows to drying.

Line 83: The material of all the wipes used in this study is not detailed and these are factors that could influence the effectiveness of disinfection because there may be interaction between the material or fiber type of the wipe and the disinfectant. This would be related to the amount of solution released to the surface.

Line 114: The authors do not specify how they adjust the bacterial and yeast suspension.

Line 117: The authors specify that they have used dirty conditions (presence of organic matter) but they do not use clean conditions (absence of organic matter) and it could be interesting because these are parameters that influence in antimicrobial efficacy. Have they carried out these experiments?

Line 121: Are there any interaction studies with disinfectants tested and the material of self made disinfectant wipes?

Have you established a protocol to ensure complete disinfection of the test surface?

Line 150: It is not indicate the equipment used to the bacterial count.

Lines 172,245,302: I recommend that the authors better justify why they did not achieve the values required for the standard in the case of Candida. It is surprising that a microorganism of the genus Candida does not survive in such short periods of time.

Line 325: The authors do not justify why the commercial wipes did not show the 5-log reduction with sodium hypochlorite. Could this be due to the presence of organic compound? 

Line 345: The authors do not discuss the reason why with E.coli 1043 the 5-log reduction is not obtained.

The results obtained for the product containing didecidimethylammonium chloride and ethylenedia minetetraacetic acid are also not justified.

Line 364: They should specify for which compound that better activity has been seen in self-made wipes.

Line 366-376:

-The authors conclude that self-made wipes eliminate microorganisms more efficiently than commercial ones but we do not know some factors that could be involved, such as the material of some wipes or the interaction between the disinfectant and the textile substrate. They also do not indicate for which product.

-In addition, it is difficult to discriminate between the microbicide activity resulting from the action of the disinfectant and the wipes that could retain the microorganism by mere mechanical action.

-In line 374 it says “The method described in the EN 16615 standard, due to 374 the low recovery of C. albicans cells from the dried surface, does not allow for proper test-375 ing of the yeasticidal activity”. I consider that further experiments are needed to conclude this.

Author Response

Author's Reply to the Review Report (Reviewer 1)

The abstract should clearly indicate the novelty of the work.

The Abstract was corrected, the following sentenses were included the text:  

For the first time, using this standard, the tested  products in the form of commercial disinfectant wipes were compared with self-made wipes soaked in respective disinfectant liquids. The disinfected surfaces were simulated by homogeneous polyvinyl chloride plates. The studies were carried out not only with the  standard but also with clinical multidrug resistant microbial strains.

 In the Introduction section you should indicate some examples of medical equipments which this wipes can be applied and in which hospital equipment or surface PVC would be present.

The following part of text was introduced into Introduction:

Wipes soaked in disinfectants are used for cleaning and disinfecting surfaces of different medical devices and equipment, e.g. ultrasound heads, surfaces of beds, treatment chairs, operating tables as well as fragments of floors and walls, if necessary.

Line 35: The reference used is from 1991 and a more current one should be used.

The reference from 1991 was replaced by [2]: Suleyman, G.; Alangaden, G.; Bardossy, A.C. The Role of Environmental Contamination in the Transmission of Nosocomial Pathogens and Healthcare-Associated Infections. Curr Infect Dis Rep. 2018, 20 (12): https://doi.org/10.1007/s11908-018-0620-2.

In line 39 it says “Candida albicans, can survive for several months on dry environmental surfaces in hospitals and health care facilities despite cleaning and disinfection”. This does not agree with the results obtained for Candida albicans in this study. The authors have not been able to evaluate the yeasticidal activity, justifying that the survival in PVC was low due to the sensitivity it shows to drying.

In the publication: Kramer, A.; Schwebke, I.; Kampf. G. How long do nosocomial pathogens persist on inanimate surfaces? A systematic re-view. BMC Infect. Dis. 2006, 6:130, the  authors presented the behavior of various clinical strains. In our research, a collection strain of Candida albicans ATCC 10231, recommended for this study by European Standard, was used. Its sensitivity to drying out may be other than clinical strains.

Line 83: The material of all the wipes used in this study is not detailed and these are factors that could influence the effectiveness of disinfection because there may be interaction between the material or fiber type of the wipe and the disinfectant. This would be related to the amount of solution released to the surface.

We agree with the Reviewer's remark, but the subject of the research was not to compare the materials of which the wipes were made - this was the subject of another publication (Song, X.; Vossebein, L.; Zille, A. Efficacy of disinfectant-impregnated wipes used for surface disinfection in hospitals: a review. Antimicrob. Resist. Infect. Control. 2019, Aug 19;8:139), but to assess the suitability of the EN 16615 for testing different wipes soaked in a various disinfectants. In the variant of testing the usage of commercial - ready-to-use wipes, the determination the type of textile material is out of our research scope. In the tests carried out in accordance with the EN 16615, wipes recommended by the this standard, of composition: 55% pulp, 45% polyethylenterephthalat (PET), were used.

Line 114: The authors do not specify how they adjust the bacterial and yeast suspension.

The following sentence was introduced:

The density of microorganisms suspensions was estimated by DEN-1B McFarland Densitometer, Grant Instruments Ltd., England

Line 117: The authors specify that they have used dirty conditions (presence of organic matter) but they do not use clean conditions (absence of organic matter) and it could be interesting because these are parameters that influence in antimicrobial efficacy. Have they carried out these experiments?

We focused on conducting research in dirty conditions (3 g/L bovine albumin solution and 3 mL/L of sheep erythrocytes). This environment is more difficult for the effective operation of disinfectants. Then there is a large contamination with organic substances that can inhibit the biocidal activity. If a given preparation is effective according to the standard in dirty conditions, its activity in clean conditions is also effective. The reverse may not be the case.

Clean conditions mean not the absence of organic substances, but 10 times lower than dirty conditions, the concentration of bovine albumin.

Line 121: Are there any interaction studies with disinfectants tested and the material of self made disinfectant wipes?

There is review article by (6) Song, X .; Vossebein, L .; Zille, A. Efficacy of disinfectant-impregnated wipes used for surface disinfection in hospitals: a review. Antimicrob. Resist. Infect. Control. 2019, Aug 19; 8: 139, the authors presented the pros and cons of using wipes soaked in disinfectants from various chemical groups, such as: alcohols, chlorine and chlorine compounds, peroxides or QAC. Only for QAC, the possibility of adsorption of the active substance by cotton was identified as disadvantages, influencing the disinfection process.

There is also  publication (but in Polish language): Tarka P., Kanecki K., Tomasiewicz K.: Evaluation of chemical agents intended for surface disinfection with the use of carrier methods. Bactericidal, yeasticidal and sporocidal activity. Post. Mikrobiol., 2016, 55, 1, 99–104 (http://www.pm.microbiology.pl) [20]. Authors describe the adsorption of benzalkonium chloride only by various wipe materials such as cellulose-polyester, viscose, cellulose and polyester blend, viscose and polyester.

We cite both articles.

Have you established a protocol to ensure complete disinfection of the test surface?

The test was carried out exactly in accordance with EN 16615, which does not include testing of complete surface disinfection.

Line 150: It is not indicate the equipment used to the bacterial count.

It was introduced in the text: The colony forming units were counted with the usage of Colony Counter apparatus, IUL Instruments, Spain.

Lines 172,245,302: I recommend that the authors better justify why they did not achieve the values required for the standard in the case of Candida. It is surprising that a microorganism of the genus Candida does not survive in such short periods of time.

It may be possible that Candida cells do not adhere sufficiently - permanently to the PCV surface and cells are removed during the process of rubbing the surface. Such situation makes results difficult to interpret.

Line 325: The authors do not justify why the commercial wipes did not show the 5-log reduction with sodium hypochlorite. Could this be due to the presence of organic compound? 

The following explanation was introduced onto text:

Lack of microbial reduction at the level required by the EN 16615  (log10 reduction of ≥ 5) after using wipes soaked in sodium hypochlorite (No. 11), may be due to the following reasons: the presence of high concentration of organic substances (3 g / L bovine albumin solution and 3 mL / L of sheep erythrocytes), relatively small amount of soaking liquid remaining on the disinfected surface (0.40 ± 0.15 g) and a short contact time – 1 min, declared by the manufacturer.

Line 345: The authors do not discuss the reason why with E.coli 1043 the 5-log reduction is not obtained.

The following explanation was introduced onto text:

The reason for the lack of the bactericidal activity required by the EN 16615 standard against the clinical strain of E. coli 1043, may be explained by lower activity of QAC against Gram-negative  bacteria than against Gram-positive bacteria [6] [Song, X.; Vossebein, L.; Zille, A. Efficacy of disinfectant-impregnated wipes used for surface disinfection in hospitals: a review. Antimicrob. Resist. Infect. Control. 2019, Aug 19;8:139]. The adsorption of some amount of active substance by the wipe material may be also possible, as was described by Tarka P., Kanecki K., Tomasiewicz K.: Evaluation of chemical agents intended for surface disinfection with the use of carrier methods. Bactericidal, yeasticidal and sporocidal activity. Post. Mikrobiol., 2016, 55, 1, 99–104 [20] (http://www.pm.microbiology.pl).

The results obtained for the product containing didecidimethylammonium chloride and ethylenediaminetetraacetic acid are also not justified.

The following explanation was introduced onto text:

The lack of the activity of CW wipes soaked with a mixture of QAC and EDTA (No. 3) required by the EN 16615 standard against 3 out of 7 tested strains can be justified by the possibility of interaction of the active substance with the tissue material and the presence of organic substances.

Line 364: They should specify for which compound that better activity has been seen in self-made wipes.

In the group of SMW wipes, only those soaked with the QAC and EDTA mixture do not present activity in accordance with the EN 16615 against 3 out of 7 tested strains of bacteria. SMW wipes soaked with the remaining tested substances show a reduction of microorganisms at the level required by the EN 16615. According to this standard, when no microbial growth is observed in the test trials, the reduction degree is only defined as a value above 5 logs and the number over 5 depends on the initial density of the microbial suspension, so it is impossible to justified which product is better – showing higher biocidal activity, it is only known that products meet the standard requirements.

Line 366-376:

The authors conclude that self-made wipes eliminate microorganisms more efficiently than commercial ones but we do not know some factors that could be involved, such as the material of some wipes or the interaction between the disinfectant and the textile substrate. They also do not indicate for which product.

The following explanation was introduced onto text:

The differences in the bactericidal activity of SMW and CW wipes may be also due to the amount of soaking fluid (products No. 1 and No. 7).

In addition, it is difficult to discriminate between the microbicide activity resulting from the action of the disinfectant and the wipes that could retain the microorganism by mere mechanical action.

The disinfectant product meets the requirements of EN 16615 if the bacteria count reduction degree on the inoculated field (No. 1) is at least 5 logs and the average number of cells in fields 2-4 does not exceed 50 cfu, and therefore the possibility of living microorganisms remaining as a result of rubbing is taken into account. However, a cells count reduction on inoculated field No. 1 must be fulfilled.

In line 374 it says “The method described in the EN 16615 standard, due to 374 the low recovery of C. albicans cells from the dried surface, does not allow for proper test-375 ing of the yeasticidal activity”. I consider that further experiments are needed to conclude this.

The text was changed as follow:

The assessment of the suitability of EN 16615 for determining the biocidal activity of disinfection products with the use of the C. albicans strain requires further research by the authors of the standard, taking into account e.g. the modification or change of the carrier surface simulating the conditions of product use.  Adherence of Candida cells to the PVC surface is much worse than adherence of bacterial cells.

Reviewer 2 Report

The authors describe an important point in disinfection. I only have two comments.

Abstract is a little confuse in L16 because, it is not specified if those results were obtained with commercial or self-made wipes. 

Authors used different evaluated solutions as a single solutions or as commercial wipes, it could be helpful if authors describe if some solutions were evaluated as commercial or self-made wipes. For example, group 1 and 7 are the same as well all groups 2 and 8 or 4 and 10. If authors agree, it will need more discussion about those comparations. 

Author Response

Author's Reply to the Review Report (Reviewer 2)

Abstract is a little confuse in L16 because, it is not specified if those results were obtained with commercial or self-made wipes. 

The abstract was chaged and the following text was introduced:

For the first time, using this standard, the tested  products in the form of commercial disinfectant wipes were compared with self-made wipes soaked in respective disinfectant liquids. The disinfected surfaces were simulated by homogeneous polyvinyl chloride plates. The studies were carried out not only with the  standard but also with clinical multidrug resistant microbial strains.

Authors used different evaluated solutions as a single solutions or as commercial wipes, it could be helpful if authors describe if some solutions were evaluated as commercial or self-made wipes. For example, group 1 and 7 are the same as well all groups 2 and 8 or 4 and 10. If authors agree, it will need more discussion about those comparations. 

Table 1 contains data concerning solutions prepared for soaking wipes, these are products 1-6. Ready, commercial wipes are products 7-12. A comparison of the antimicrobial activities of SMW and CW wipes containing the same active ingredients is presented in Figure 3.

Reviewer 3 Report

-Authors should clearly specify what the objectives of the study are, as it is unclear.

-Why did the authors only run the test 3 times? Are they enough?

-There are significant differences between the use of commercial wipes and soak wipes? 

-Does the study have limitations?

-Can you come to this conclusion "Self-made wipes killed microorganisms more efficiently than commercial 370 wipes... (line 370)" without running the statistics?

Author Response

Author's Reply to the Review Report (Reviewer 3)

Authors should clearly specify what the objectives of the study are, as it is unclear.

The following text was introduced on the end of the Introduction:

The aim of our work was the practical application of a relatively new European Standard dedicated for testing disinfectants using a carrier in the form of PVC plates and with the mechanical action. The methods used so far did not allow for testing products such as commercially available, ready-made wipes or self-soaked wipes under simulated conditions of their use.

Why did the authors only run the test 3 times? Are they enough?

According to EN 16615, which refers to the number of repetitions to EN 14885, three repetitions of tests are sufficient.

Moreover, our standard deviation results were low, from 0.029 to 0.507.

There are significant differences between the use of commercial wipes and soak wipes? 

According to EN 16615, when no growth is observed on the PVC field No 1, the degree of cells count reduction is defined as a value above 5 logs. The reduction degree depends on the initial density of the microorganisms suspension, so it is not possible to say which product is better, it is only known that  it meets the requirements of the standard.

Does the study have limitations?

The lack of a suitable neutralizer for the active substance used in soaking wipes mixture may be a crucial factor preventing the testing according to EN 16615. The inability to inhibit the antimicrobial action of the product against the tested strains does not allow for the determination of the duration of action. In our work, the suitability of appropriate neutralizers was tested.

Can you come to this conclusion "Self-made wipes killed microorganisms more efficiently than commercial 370 wipes... (line 370)" without running the statistics?

According to EN 16615, when no growth is observed on the PVC field No. 1, the degree of cells count reduction is defined as a value above 5 logs. The reduction degree depends on the initial density of the microorganisms suspension, so it is not possible to say which product is better, it is only known that  it meets the requirements of the standard.

One of the conclusions of our work is the statement that among the tested wipes, more of those meeting the requirements of the standard were among SMW wipes than CW.

Round 2

Reviewer 1 Report

I consider that the authors have made a thorough review of the comments and have made substantial improvements to the manuscript. After the corrections and based on the justifications provided by the authors, I conclude that the manuscript may be of great interest to the readers of the journal.